# Serum Interleukin-36 α as a Candidate Biomarker to Distinguish Behçet’s Syndrome and Psoriatic Arthritis

**DOI:** 10.3390/ijms24108817

**Published:** 2023-05-16

**Authors:** Alessandra Bettiol, Filippo Fagni, Irene Mattioli, Giacomo Bagni, Gianfranco Vitiello, Alessia Grassi, Chiara Della Bella, Marisa Benagiano, Arianna Troilo, Katarzyna Stella Holownia, David Simon, Flavia Rita Argento, Jurgen Sota, Claudia Fabiani, Matteo Becatti, Claudia Fiorillo, Georg Schett, Giuseppe Lopalco, Luca Cantarini, Domenico Prisco, Elena Silvestri, Giacomo Emmi, Mario Milco D’Elios

**Affiliations:** 1Department of Experimental and Clinical Medicine, University of Firenze, 50134 Firenze, Italy; alessandra.bettiol@unifi.it (A.B.); irene.mattioli@unifi.it (I.M.); giacomo.bagni@unifi.it (G.B.); vit.gianfranco@gmail.com (G.V.); alessia.grassi@unifi.it (A.G.); chiara.dellabella@unifi.it (C.D.B.); marisa.benagiano@unifi.it (M.B.); arianna.troilo@unifi.it (A.T.); hanorah@gmail.com (K.S.H.); domenico.prisco@unifi.it (D.P.); elena.silvestri@unifi.it (E.S.); 2Department of Internal Medicine 3—Rheumatology and Immunology, Friedrich Alexander University Erlangen-Nuremberg and Universitätsklinikum Erlangen, 91054 Erlangen, Germany; filippo.fagni@uk-erlangen.de (F.F.); david.simon@uk-erlangen.de (D.S.); georg.schett@uk-erlangen.de (G.S.); 3Department of Experimental and Clinical Biomedical Sciences “Mario Serio”, University of Firenze, 50134 Firenze, Italy; flaviarita.argento@unifi.it (F.R.A.); matteo.becatti@unifi.it (M.B.); claudia.fiorillo@unifi.it (C.F.); 4Department of Medical Sciences, Surgery and Neurosciences, Research Center of Systemic Autoinflammatory Diseases and Behçet’s Disease Clinic, University of Siena, 53100 Siena, Italy; sota@student.unisi.it (J.S.); cantarini@unisi.it (L.C.); 5Ophthalmology Unit, Department of Medicine, Surgery and Neurosciences, University of Siena, 53100 Siena, Italy; claudia.fabiani@gmail.com; 6Rheumatology Unit, Department of Emergency and Organs Transplantation (DETO), University of Bari, 70124 Bari, Italy; glopalco@hotmail.it; 7Centre for Inflammatory Diseases, Monash University Department of Medicine, Monash Medical Centre, Clayton, VIC 3168, Australia; 8Department of Molecular and Developmental Medicine, University of Siena, 53100 Siena, Italy

**Keywords:** Behçet disease, biomarkers, cytokines, interleukin 36, vasculitis

## Abstract

Behçet’s syndrome (BS) is a rare systemic vasculitis characterized by different clinical manifestations. As no specific laboratory tests exist, the diagnosis relies on clinical criteria, and the differential diagnosis with other inflammatory diseases can be challenging. Indeed, in a relatively small proportion of patients, BS symptoms include only mucocutaneous, articular, gastrointestinal, and non-typical ocular manifestations, which are frequently found also in psoriatic arthritis (PsA). We investigate the ability of serum interleukin (IL)-36α—a pro-inflammatory cytokine involved in cutaneous and articular inflammatory diseases—to differentiate BS from PsA. A cross-sectional study was performed on 90 patients with BS, 80 with PsA and 80 healthy controls. Significantly lower IL-36α concentrations were found in patients with BS as compared to PsA, although in both groups IL-36α was significantly increased compared to healthy controls. An empirical cut-off of 420.6 pg/mL displayed a specificity of 0.93, with a sensitivity of 0.70 (AUC 0.82) in discriminating PsA from BS. This cut-off displayed a good diagnostic performance also in BS patients lacking highly specific BS manifestations. Our results indicate that IL-36α might be involved in the pathogenesis of both BS and PsA, and might be a candidate biomarker to support the differential diagnosis of BS.

## 1. Introduction

Behçet’s syndrome (BS) is a rare chronic systemic vasculitis hallmarked by oral and genital ulcerations and ocular involvement. Among other disease manifestations, articular, cutaneous, vascular, neurological, and gastrointestinal involvement is also frequent in BS [1,2,3].

Differently from cases presenting with peculiar pathognomonic manifestations (such as genital ulcers, posterior uveitis, or vascular events), differential diagnosis of BS patients reporting mucocutaneous, articular, anterior ocular and/or gastrointestinal manifestations in the absence of other major organ involvement can be challenging [4]. 

Articular involvement is reported in up to 80% of BS patients of both sexes [2,5,6,7] and may resemble seronegative arthritis, particularly psoriatic arthritis (PsA), with which an overlap has been described as both diseases can be ascribed to MHC-I-opathies [8,9,10].

The diagnostic process can be challenging in the case of patients with PsA *sine psoriasis* (i.e., in the absence of psoriatic skin lesions), accounting up to 20% of the cases of PsA [11]. Moreover, extra-articular manifestations, including gastrointestinal, anterior ocular, cardiovascular, and non-psoriatic cutaneous involvement, might be found in up to 35% of patients with PsA and in up to 50% of BS patients, further complicating the differential diagnosis [2,12,13,14]. Notably, common immunopathogenic pathways, including among others key cytokines such as tumor necrosis factor (TNF) alpha, interleukin (IL-17), IL12 and IL23, sustain articular and extra-articular manifestations in both diseases [1,6,15,16,17]. The relationship between BS and PsA is also supported by the recent evidence that BS patients present a significantly increased risk of psoriasis, and still greater risk of PsA, being twice more vulnerable to PsA as compared to non-BS subjects [13].

Differentiating BS from PsA, especially in low incidence areas for BS, can be relevant, from a clinical point of view. However, to this date no specific instrumental or laboratory biomarker is available to help the differential diagnosis between these two conditions, which currently relies only on clinical assessment [18].

IL-36 (including the three agonists IL-36α, β, γ, and the inhibitor IL-36Ra) belongs to the IL-1 family involved in innate immune responses [19]. The IL-36 axis has been associated with skin and joint-related inflammatory conditions, as it stimulates the production of pro-inflammatory mediators by synovial fibroblasts [20,21]. Moreover, aberrant IL-1 signalling, and IL-1-related gene polymorphisms are implicated in the pathogenesis of BS, and anti-IL-1 therapies have shown some efficacy in the treatment of refractory BS [22,23,24,25,26]. Accordingly, increased serum levels of IL-36, especially IL-36α [27,28], have been described in psoriasis [29] as well as in PsA at synovial level [30], although no study investigated its levels in BS.

Interestingly, IL-36 family members need to be processed to acquire their fully active form [31,32]. Neutrophil proteases (particularly neutrophil elastase) have been identified as the main regulators of this processing, and neutrophil extracellular traps (NETs) can act as a platform for IL-36 activation [33]. Neutrophils are known to play a central role in BS pathogenesis, also via NETs release [34], whereas NETosis has been more rarely reported in PsA [35]. On these bases, in this study we assessed serum IL-36α levels in a cohort of BS patients, PsA and healthy controls, and investigated the ability of serum IL-36α to differentiate patients with BS from those with PsA.

## 2. Results

### 2.1. Patients Characteristics

A total cohort of 90 BS patients was included and was compared to 80 patients diagnosed with PsA and 80 healthy controls (HCs). Demographic, clinical, and therapeutic features of the cohorts are summarized in Table 1.

In the BS cohort, 57% (*n* = 51) of patients were men, with a median age at time of inclusion in the study of 45 (IQR 36–55) years. Regarding disease manifestations in the whole medical history, 93% (*n* = 84) of patients had history of oral ulcers, with 38% also having genital aphthosis (*n* = 34). Among other common manifestations, cutaneous involvement was reported in 81% (*n* = 73), articular manifestations in 52% (*n* = 47), gastrointestinal symptoms in 38% (*n* = 34), uveitis in 34% (*n* = 31), vascular involvement in 19% (*n* = 17), and neurological manifestations in 18% (*n* = 16). A total of 44% of patients (*n* = 40) were HLA-B51 positive. At time of inclusion in the study, half of the patients (*n* = 44) were classified as having active disease (defined by a Behçet’s Disease Current Activity Form (BDCAF) ≥ 1), the median BDCAF in the BS cohort being of 0 (0–3). Regarding ongoing immunomodulating treatment, most patients were receiving conventional and/or biologic (cs/b) disease-modifying antirheumatic drugs (DMARDs) (32% and 28%, *n* = 29 and 25, respectively), whereas 40% (*n* = 36) were on corticosteroid monotherapy or in combination (≥5 mg prednisone daily), or colchicine monotherapy.

Within the BS cohort, eight patients presented without major BS-specific organ manifestations. All of them fulfilled the ICBD criteria. Their clinical and therapeutic features are summarised in Table A1.

In the PsA and HC cohorts, 53% (*n* = 42 in each group) of patients were men, with a median age at inclusion of 50 (38–66) years and 45 (38–52) years, respectively. In the PsA group, all patients had a history of arthritis and 90% (*n* = 72) suffered from psoriatic skin disease, meaning 10% of PsA participants had a PsA *sine psoriasis*. A history of uveitis was present in 6.3% (*n* = 5) of patients, and gastrointestinal symptoms were present in 11% (*n* = 9). Deep vein thrombosis was reported in 7 (9%) of patients. At the time of inclusion 54% (*n* = 43) of PsA patients had achieved MDA, the median PASI score being 1.2 (IQR: 0.0–2.1). The majority of PsA patients (46%, *n* = 37) were receiving treatment with bDMARDs and 39% (*n* = 30) with csDMARDs, whereas 13% (*n* = 10) were only receiving symptomatic treatment with NSAIDs. Only three patients (4%) were receiving ≥5 mg of prednisone daily. An overview of the clinical features of PsA controls is reported in Table 1.

### 2.2. IL-36α Levels in Patients with BS, PsA, and in HCs

The median level of IL-36α was significantly higher among BS patients (201.7 (112.7–320.2) pg/mL) as compared to HCs (16.9 (13.7–22.2); *p* < 0.001). Conversely, BS patients displayed significantly lower IL-36α levels as compared to the PsA group (544 (296–759); *p* < 0.001) (Figure 1).

We then investigated the ability of IL-36α to discriminate BS patients from PsA patients: the ROC curve analysis revealed an Area Under the ROC Curve (AUC) of 0.842 (95% CI: 0.782–0.902), indicating that there is an 84% (95% CI: 78–90%) chance that the use of IL-36α will be able to distinguish between BS and PsA patients. In particular, the empirical optimal cut-off resulted to be of 420.6 pg/mL. This cut-off displayed a specificity of 0.93, with a sensitivity of 0.70 (Figure 2), indicating that IL-36α levels above 420.6 pg/mL have a 93% change to correctly identify patients with PsA, with a 70% change of correctly discriminating people without PsA (i.e., with BS).

### 2.3. Variations in IL-36α Levels among BS Patients

Within the BS cohort, we further investigated differences in IL-36α levels according to clinical and therapeutic features (Table 2). IL-36α levels were significantly higher in patients with history of genital aphthosis (269.4 (173.2–384.5), *p* = 0.015) as compared to BS patients without this manifestation or uveitis (243.4 (192.2–345.1), *p* = 0.029). Furthermore, a positive correlation was found between IL-36α levels and BDCAF at time of sample collection (Spearman’s rho = 0.492; *p* < 0.001), although no significant difference emerged between patients with active vs. inactive activity. Conversely, IL-36α levels did not significantly vary according to the other BS disease manifestations in the whole medical history or HLA-B51 positivity. Moreover, no significant difference in IL-36 levels emerged between patients receiving corticosteroids/colchicine alone, csDMARDs or biologic DMARDs. As IL-36α levels were found to be remarkably increased in BS patients with history of genital or ocular involvement, subgroup analyses were conducted to assess whether IL-36α was able to also discriminate patients with these manifestations from those diagnosed with PsA. IL-36α displayed an AUC of 0.796 (95% CI: 0.715–0.876) in discriminating BS patients with genital aphthosis from those with PsA, and an AUC of 0.798 (95% CI: 0.719–0.879) in discriminating BS patients with uveitis from those with PsA. 

In a second subgroup analysis, we focused on the eight patients with BS without major organ involvement (i.e., without vascular, neurological, or posterior ocular manifestations), as this can represents a challenging group from a diagnostic point of view.

Notably, all eight patients had IL-36α levels below the cut-off value 420.6 pg/mL, ranging between 155.9 and 387.5 pg/mL. For this subgroup of patients, the AUC as compared to the PsA cohort was of 0.827 (95% CI: 0.732–0.918).

## 3. Discussion

This study shows for the first time that serum IL-36α is increased in patients with PsA as well as in BS, although to a lesser extent, and indicates that serum IL-36α could be a candidate biomarker for the differential diagnosis between these two conditions.

Posing an accurate diagnosis of BS is crucial for establishing a proper treatment strategy and predicting disease course and prognosis. As there are no pathognomonic laboratory tests, the diagnosis relies on clinical criteria. This is particularly true for patients lacking more BS-specific manifestation, such as genital ulcers, posterior uveitis, neurological, or vascular involvement [4]. Indeed, in a relatively small proportion of patients, BS symptoms may only include mucosal, articular, gastrointestinal, cutaneous, and anterior ocular manifestations, which are frequently found also in patients with PsA. Accordingly, patients might meet both ICBD and CASPAR criteria for BS and PsA, which further complicate accurate diagnosis.

On these bases, we assessed serum IL-36α as a potential laboratory biomarker for differential diagnosis between BS and PsA.

IL-36α, as the other members of the IL-1 family, play a central role in innate immunity as well as in the pathogenesis of immune-mediated disorders [19]. Polymorphisms in the IL-1 gene cluster (namely IL-1A 2889C and IL-1B +5887T haplotype) are associated with an increased susceptibility to vasculitis, including BS, and other cytokines belonging to the IL-1 family (including IL-1α and IL-33) are known to be involved in the pathogenesis of BS [36,37,38], whereas little is known about the role of IL-36 in this disease.

Conversely, increased levels of IL-36 have been consistently reported in the skin and joint of patients with psoriasis [29] and PsA [30], where it mediates cutaneous and synovial inflammation, and drugs targeting the IL-36 axis are under investigation for the treatment of psoriasis [39,40].

The results from this study indicate that patients with BS present significantly lower serum IL-36α levels as compared to those with PsA, and that IL-36α can effectively differentiate between the two conditions. Particularly, an empirical cut-off of 420.6 pg/mL is associated with a very high specificity (0.93), paired by a good sensitivity (0.70).

On their turn, our findings indicate that BS patients displayed significantly higher IL-36α levels as compared to HCs. Particularly, patients with mucocutaneous (ocular and genital) aphthosis and patients with uveitis presented the highest IL-36α levels.

The biological interpretation of these findings is debatable. The IL-36 axis is known to be involved in mucosal repair; indeed, recent studies showed that the IL-36 axis promotes IL-23/IL-22-dependent mucosal healing, and mice lacking IL-36 present impaired recovery from mucosal damage [34,41,42]. Thus, we can speculate that the higher levels of IL-36 found in patients with oral and genital aphthosis might be the result of a protective “repairing” mechanism in response to the damaged barrier function to epithelial and mucosal tissue, which is also a common feature to psoriasis.

Interestingly, in our BS cohort serum IL-36α levels were remarkably higher in the subgroup of patients with mucocutaneous and ocular involvement, although beyond the cut-off of 420 pg/mL. Accordingly, this cytokine still effectively discriminated patients with BS with manifestations associated with the highest IL-36α levels (i.e., with mucocutaneous and ocular involvement) from those diagnosed with a PsA, with a good AUC (around 0.80). However, from a clinical point of view, using IL-36α as a diagnostic biomarker in BS patients with mucocutaneous or ocular involvement is not necessary, as most mucocutaneous manifestations (such as genital ulcers) or ocular ones (such as posterior uveitis) are almost pathognomonic of BS [4].

Concomitantly, we found that also in patients with BS lacking major disease-specific organ involvement, serum IL-36α levels were far beyond the cut-off of 420.6 pg/mL which discriminates BS from PsA. From a clinical point-of-view, these findings indicate a potential clinical applicability of IL-36, as a diagnostic biomarker to assist physicians in the diagnosis of challenging cases.

Notably, the fact that IL-36α levels were not increased in patients with cutaneous or articular BS manifestations might suggest that different pathogenetic mechanisms sustain these manifestations in this condition.

Some limitations should be considered when interpreting these results. First, our study includes relatively small cohorts and investigates one single isoform of IL-36. The choice of focusing on the isoform of IL-36α was driven by the fact that a body of the literature reports on the increased levels of this isoform in PsA [43]. Moreover, IL-36α is known to be highly abundant in epithelial cells, as well as monocytes, B cells, and T cells, and it has an important role in skin barrier function and homeostasis [44,45]. As mucocutaneous manifestations are frequent in BS and a disrupted skin-barrier function (e.g., of gut barrier) has been suggested in this condition [46], we decided to focus the analysis on this isoform. Regarding the cellular sources of this cytokine, IL-36α is known to be expressed by a variety of cells, including keratinocytes, T and B lymphocytes, monocytes, dermal fibroblasts and endothelial cells [44]. In PsA, CD138-positive plasma cells are known to be the main cellular source of IL-36α [43]. As for BS, we might speculate that endothelial cells could be a relevant source of IL-36α, as endothelial activation and chronic endothelial damage are known to be a major pathogenetic feature in BS [3,47]. However, this was not investigated in the present work and should be explored in future studies. As such, the relationship of BS and PsA with other IL36 family members remains to be determined. Second, no validation analysis was conducted, and external validation of this biomarker is required. Third, BS patients with clinical features resembling with PsA were around 10% of the total cohort; thus, the performance of this possible biomarker in a group of patients with features representing a real diagnostic challenge should be further investigated. Fourth, given the cross-sectional study design, the influence of background therapies on IL-36 levels could not be assessed. However, considering the higher proportion of patients treated with biologics in the cohort with PsA as compared to that with BS, we could speculate that in untreated patients the difference in IL-36 levels between the two diseases could be even greater. Fifth, patients were not enrolled at time of diagnosis, which would be the optimal timepoint to assess a diagnostic biomarker, and the impact of pharmacological therapies or clinical manifestations developing during follow-up on IL-36 levels could not be excluded. Finally, as IL-36 is not routinely assessed in clinical practice, the applicability of this test in the routine diagnostic approach remains to be explored.

## 4. Materials and Methods

### 4.1. Study Design and Population

A cross-sectional study was performed in a cohort of adult BS, PsA patients and HCs (Figure A1). BS patients were followed at two referral centres for BS (Behçet Center of the Careggi University Hospital of Florence, and University Hospital of Siena, Italy) and met the ICBD diagnostic criteria for BS [8].

BS patients were compared with a control group of patients diagnosed with PsA from the University Hospital Erlangen (Erlangen, Germany) who met the CASPAR criteria for PsA [14].

A HC group was recruited through an open campaign at the Careggi University Hospital of Florence. Exclusion criteria for the HC group included any diagnosis of immune-mediated inflammatory disease (e.g., rheumatoid arthritis, systemic lupus erythematosus, etc.), active infections, history of cerebro- and/or cardiovascular diseases or cancer.

### 4.2. Data and Sample Collection

Demographic information including age and sex were collected from all study participants and their distribution was comparable within HCs and patients with BS and PsA. For BS and PsA patients, clinical and therapeutic data were collected. The presence of recurrent oral and genital aphthosis, arthritis, inflammatory gastrointestinal symptoms (e.g., chronic abdominal pain, bloody and/or mucous diarrhoea), uveitis, and vascular manifestations defined as vasculitis and/or occurrence of venous or arterial thrombosis was reported. Psoriatic skin disease presence and severity measured by the Psoriasis Area Severity Index (PASI) was scored in PsA patients only, whereas for BS patients, we reported the presence or absence of typical BS skin manifestation such as ulcers and erythema nodosum. The presence of neurological manifestations defined as central nervous system inflammatory lesions was also reported in BS patients.

For BS patients, disease activity at time of enrolment was assessed by means of the BDCAF, and patients with a BDCAF ≥ 1 were defined as having active disease. Conversely, in PsA patients, the presence of active disease was defined using the minimal disease activity (MDA) score. If MDA was not achieved, PsA was considered active.

Data about ongoing immunomodulatory therapies at the time of sampling were also collected for all patients, including corticosteroids, colchicine, conventional csDMARDs biologic DMARDs, and non-steroidal anti-inflammatory drugs (NSAIDs). All subjects provided written informed consent for the inclusion in the study. The study was approved by the Ethic Committee Area Vasta Centro—Careggi (Ref. CEAVC 12804; date of approval 13/11/2018). The patients/participants provided their written informed consent to participate in this study and was conducted in accordance with the ethical principles of the Declaration of Helsinki.

### 4.3. IL-36α Quantification

Blood samples from BS and PSA patients and HCs were collected by venipuncture of antecubital vein, and placed into vacutainers (BD Vacutainer Systems, Plymouth, UK) containing ethylenediaminetetraacetate 0.17 mol/L. Serum IL-36α concentrations were measured at the immunology laboratory of the Department of Experimental and Clinical Medicine, University of Firenze (Firenze) using human IL-36α enzyme-linked immunosorbent assay (ELISA) kits (MyBioSource, San Diego, CA, USA) according to the manufacturer’s recommendations. All experiments were performed in duplicate.

### 4.4. Statistical Analysis

Continuous variables were reported as median values and interquartile range (IQR) and compared between BS and PsA patients and between BS patients and HCs using the Mann–Whiney test for the unpaired. Categorical variables were reported as absolute numbers and percentages. Within the BS cohort, variations in IL-36α levels according to clinical features (disease manifestations and disease activity) and to treatment, were also assessed. Furthermore, correlation between IL-36α levels and BDCAF at time of sample collection was assessed using the Spearman’s R test.

Logistic regression models were fitted, and Receiving Operating Characteristics (ROC) were derived to assess the AUC of IL-36α in discriminating BS and PsA patients. Empirical estimation of the optimal cut-point for IL36α as a possible diagnostic test was computed using the Youden method. Statistical significance was considered for *p*-values < 0.05.

The sample size for the BS and PsA cohorts was defined based on the feasibility of patient enrolment during the study period, also considering the rareness of BS. The *post hoc* power calculation, which was performed considering a BS cohort of 90 patients and a PsA cohort of 80 patients, with an AUC of 0.842, revealed a power >99%, with an alpha-error of 5%.

In a subgroup analysis, the ability of IL36α to discriminate BS patients without major disease-specific organ manifestations (i.e., presenting mucocutaneous, articular, anterior non-granulomatous uveitis, or gastrointestinal involvement without vascular or neurological disease manifestation) from PsA patients was assessed, as differential clinical diagnosis in this setting can be challenging.

## 5. Conclusions

Taken together, our results indicate for the first time that serum IL-36α is remarkably increased in patients with PsA as well as in BS, although to a lesser extent, and suggest that it could be a candidate biomarker for the differential diagnosis between these two conditions. Particularly, IL-36α levels exceeding the cut-off of 420 pg/mL seem highly specific for PsA; thus, in patients with a clinical presentation resembling both BS and PsA, high IL-36 levels might be used to exclude a clinical suspicion of BS. In the absence of an established biomarker for both BS and PsA, these results can represent the starting point for future studies aimed at assessing the role of a combination of multiple biomarkers for better understanding the pathophysiology of these inflammatory diseases, and for diagnostic purposes as well.

## Figures and Tables

**Figure 1 ijms-24-08817-f001:**
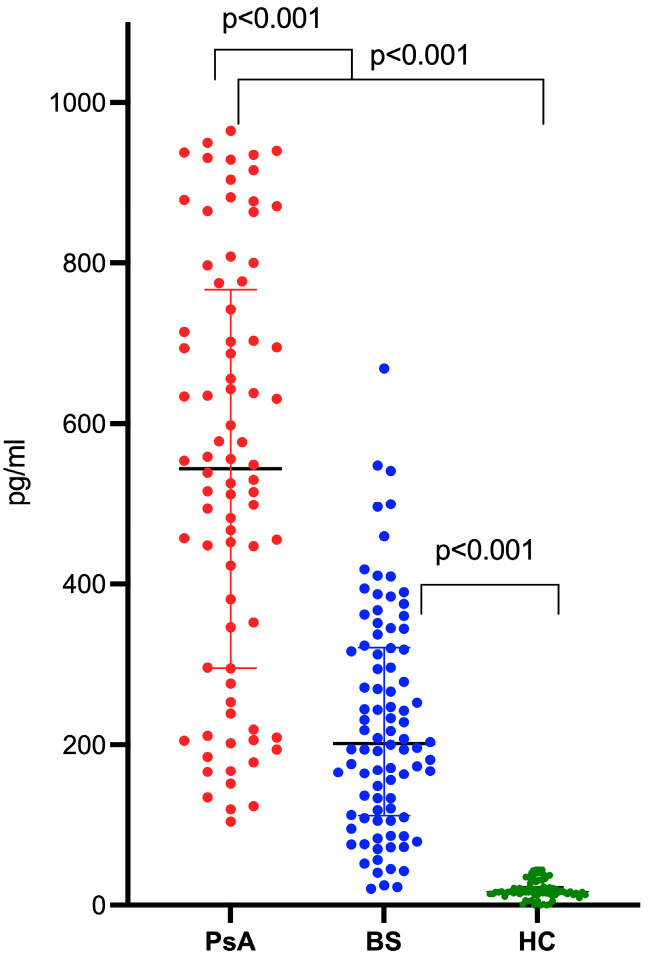
IL-36α levels in patients with Behçet’s syndrome (BS), with psoriatic arthritis (PsA), and in healthy controls (HCs).

**Figure 2 ijms-24-08817-f002:**
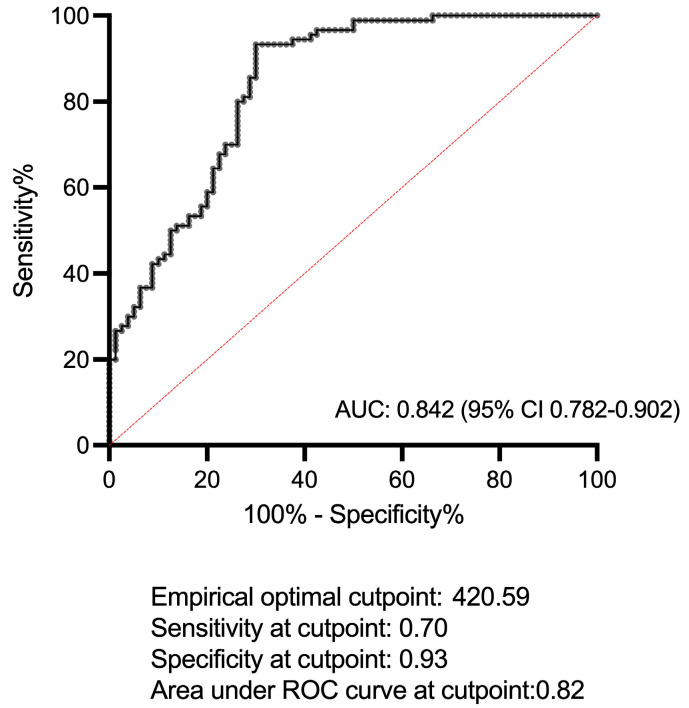
ROC curve for the sensitivity and specificity of IL-36α in discriminating patients with Behçet’s syndrome (BS) from patients with psoriatic arthritis (PsA).

**Table 1 ijms-24-08817-t001:** Demographic, clinical, and therapeutic characteristics of patients with Behçet’s syndrome (BS), of control patients with psoriatic arthritis (PsA) and of healthy controls.

	BS N (% Out of 90)	PsA N (% Out of 80)	Healthy Controls N (% Out of 80)
Sex—male	51 (56.7)	42 (52.5)	42 (52.5)
Age, years (median, IQR)	45 (36–55)	50 (38–66)	45 (38–52)
**Disease manifestations**			
Oral aphthosis	84 (93.3)	2 (2.5)	
Genital aphthosis	34 (37.8)	0 (0)	
Cutaneous	73 (81.1)	-	
Psoriasis	-	72 (90.0)	
Arthritis	47 (52.2)	80 (100.0%)	
Gastrointestinal symptoms	34 (38.2)	9 (11.3%)	
Uveitis	31 (34.4)	5 (6.3%)	
Vascular	17 (18.9)	7 (8.8%) *	
Neurological	16 (17.8)	-	
**HLA-B51 positivity**	40 (44.4)	-	
**Active disease**	44 (48.9)	37 (46.3)	
BDCAF (median, IQR)	0 (0–3)	-	
PASI (median, IQR)	-	1.2 (0–2.1)	
**Ongoing immunomodulating treatment**			
No treatment/only NSAIDs	0	10 (12.5)	
Corticosteroids/colchicine	36 (40.0)	3 (3.8)	
csDMARDs	29 (32.2)	30 (38.5)	
Methotrexate	0	28 (35.0)	
Leflunomide	0	2 (3.5)	
Biologic (±cs) DMARDs	25 (27.8)	37 (46.3)	
TNFi	21 (23.3)	27 (33.8)	
IL12/23i	0	3 (3.8)	
IL1i	4 (4.4)		

BDCAF: Behçet’s Disease Current Activity Form; BS: Behçet syndrome; csDMARD: conventional synthetic disease-modifying antirheumatic drugs; HLA: human leukocyte antigen; IQR: interquartile range; IL12/23i: interleukin 12/23 inhibitors; NSAIDs: non-steroidal anti-inflammatory drugs; PASI: psoriasis area and severity index; PsA: Psoriatic-Arthritis; TNFi: tumor necrosis factor inhibitors; IL1i: interleukin 1 inhibitors. Definitions for disease manifestations are provided in the methods section. * Patients with PsA with a history of deep vein thrombosis.

**Table 2 ijms-24-08817-t002:** Variations in IL-36α levels within the BS cohort, according to manifestations in the whole medical history, presence of HLA-B51, active disease at time of enrolment and pharmacological therapies.

	With the Considered Feature	Without the Considered Feature	*p*-Value #
	IL-36 (pg/mL)	IL-36 (pg/mL)	
Overall	201.7 (112.7–320.2)		
**Disease manifestations**			
Genital aphthosis	269.4 (173.2–384.5)	176.1 (109.2–266.3)	0.015 *
Cutaneous	207.2 (132.9–323.4)	192.2 (94.9–271.2)	0.284
Arthritis	201.7 (120.2–320.2)	194.2 (104.8–344.2)	0.784
Intestinal symptoms	212.2 (104.8–360.4)	200.2 (118.2–316.5)	0.950
Uveitis	243.4 (192.2–345.1)	173.2 (94.9–312.3)	0.029 *
Vascular	269.4 (208.2–351.4)	192.2 (108.1–296.2)	0.072
Neurological	228.3 (171.1–375.2)	194.3 (108.1–318.3)	0.240
**HLA-B51**	229.7 (152.0–340.8)	193.2 (94.9–312.3)	0.237
**Active disease**	200.7 (106.5–379.9)	201.7 (133.0–296.2)	0.529
**BDCAF**	Spearman’s rho: 0.492		<0.001 *
**Ongoing immunomodulating treatment**			
Corticosteroids/colchicine	245.7 (160.5–348.3)		0.138
csDMARDs	192.2 (120.2–252.2)		
Biologic (±cs) DMARDs	176.1 (82.9–294.4)		

# *p*-value from Mann-Whitney or Krusal–Wallis test for unpaired data, comparing IL-36α levels in patients with vs without a considered feature. * Statistically significant with *p*-value < 0.05. BDCAF: Behçet’s Disease Current Activity Form; csDMARD: conventional synthetic disease-modifying antirheumatic drugs; HLA: human leukocyte antigen.

## Data Availability

The raw data supporting the conclusions of this article will be made available by the authors, without undue reservation.

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
