# Peer review of "Serum Interleukin-36 α as a Candidate Biomarker to Distinguish Behçet’s Syndrome and Psoriatic Arthritis"

_ijms, 2023, doi:10.3390/ijms24108817_

Round 1
Reviewer 1 Report
The manuscript under review presents the results of IL-36alfa serum determinations in three cohorts of patiemts with PsA, Behçet disease, and healthy controls. The introduction is a bit too extensive, and the potential role of IL-36alfa determinations for differential diagnostic purposes with PsA seems a bit preposterous in most patients (cf doi: 10.1093/rheumatology/kead101), even though I do agree that the differential diagnosis of Behçet disease can be broad. Perhaps the authors should just comment that Behçet disease and PsA can be considered to belong in the overarching category of MHC-I-opathies (doi: 10.1136/ard-2022-222852) and they wanted to assess the serum levels of IL-36alfa in the aforementioned cohorts. The confounding role of treatment is acknowledged in the Limitations section of the manuscript Discussion. The authors should make clear where were the determinations of IL-36alpha performed, since the cohorts had different geographic origins. They should discuss why they chose IL-36alpha, and not IL-36gamma or IL-36antagonist, for instance. They should also discuss what is the putative cell source of the IL-36alfa they measured, and which potential effects could it have on which targets (endothelium in Behçet, synovium in PsA?). Potential correlation of IL-36a levels with level of activity, rather than organ involvement, should be adressed in Behçet, and especially in PsA, taking into account the wider distribution of values in the latter cohort. Incidentally, what do the authors mean by 'major organ involvement' displayed by 82 out of 90 patients? This does not seem a run of the mill Behçet disease population in Europe...
Acceptable; some minor style corrections required.
Reviewer 2 Report
Reviewer comments and suggestions
The authors in this study investigated the ability of serum interleukin (IL)-36α - a pro-inflammatory cytokine involved in cutaneous and articular inflammatory diseases -to distinguish Behçet’s syndrome (BS) from psoriatic arthritis (PsA). For this, the authors used cross-sectional design to perform on 90 patients with BS, 80 with PsA and 80 healthy controls. The study result noted that a significant lowering of IL-36α concentrations in patients with BS as compared to PsA was observed noted an elevation when compared with healthy controls. An empirical cut-off of 420.6 pg/ml displayed a specificity of 0.93, with a sensitivity of 0.70 (AUC 0.82) in discriminating PsA from BS. Hence, the author stated that IL-36α might be involved in the pathogenesis of both BS and PsA, and might be a candidate biomarker to support the differential diagnosis of BS.
Overall, the manuscript was well written. However, a few concerns/comments needed to be explained/modified.
- Line 71 DMARDs, please add the full form as it was the first time used.
- Line 73 Please highlight and explain comprehensively similar to these studies (reference 13)
- It would be nice if the authors could explain the material and methods with the help of a ray diagram. Line 115-122
- Please explain the result of ROC, one line was not sufficient.
- Table 2 please mention what was without in the legend part.
- It is important to highlight the novelty in the first para of the discussion rather than discussing the general paragraph.
- Line 232-233 what do the authors want to state here, please add relevant references if possible.
- Please add approval and ethical number for this study
- Please check the references, the format still needs to be checked again.
Round 2
Reviewer 1 Report
This reviewer's comments have been adequately addressed in the revised version